| 1           | A novel approach: community-driven snow depth measurement in                                                                                                                                                                                                                                                                                                                     |
|-------------|----------------------------------------------------------------------------------------------------------------------------------------------------------------------------------------------------------------------------------------------------------------------------------------------------------------------------------------------------------------------------------|
| 2           | Central Asia                                                                                                                                                                                                                                                                                                                                                                     |
| 3<br>4<br>5 | Abror Gafurov <sup>1,2,3,*</sup> , Anesa Sasivarevic <sup>2,6</sup> , Zulfiya Karimova <sup>2,3</sup> , Elena Grigorieva <sup>2,3</sup> , Akmal Gafurov <sup>4</sup> , Erkin Abdulakhatov <sup>4</sup> , Feruza Gafurova <sup>2</sup> , Nurmukhammad Abdurasulov <sup>2</sup> , Jakhongir Kayumbaev <sup>2</sup> Rustam Adkhamov <sup>1,6</sup> and Friedrich Busch <sup>5</sup> |
| 6           |                                                                                                                                                                                                                                                                                                                                                                                  |
| 7           | <sup>1</sup> Section 4.4: Hydrology, GFZ Helmholtz Centre for Geosciences, Potsdam, Germany                                                                                                                                                                                                                                                                                      |
| 8           | <sup>2</sup> Innovative Water & Environmental Solutions, Berlin, Germany                                                                                                                                                                                                                                                                                                         |
| 9           | <sup>3</sup> Institute of Geography, Humboldt Universität zu Berlin, Berlin, Germany                                                                                                                                                                                                                                                                                             |
| 10          | <sup>4</sup> Hydrometeorological Research Institute under Uzhydromet (NIGMI), Tashkent, Uzbekistan                                                                                                                                                                                                                                                                               |
| 11          | <sup>5</sup> Potsdam Institute for Climate Impact Research, Potsdam, Germany                                                                                                                                                                                                                                                                                                     |
| 12          | <sup>6</sup> Technical University of Berlin, Berlin, Germany                                                                                                                                                                                                                                                                                                                     |
| 13          | Corresponding author: gafurov@gfz.de                                                                                                                                                                                                                                                                                                                                             |
| 14          |                                                                                                                                                                                                                                                                                                                                                                                  |
| 15          |                                                                                                                                                                                                                                                                                                                                                                                  |
|             |                                                                                                                                                                                                                                                                                                                                                                                  |
| 16          |                                                                                                                                                                                                                                                                                                                                                                                  |
| 17          |                                                                                                                                                                                                                                                                                                                                                                                  |
| 10          |                                                                                                                                                                                                                                                                                                                                                                                  |
| 18          |                                                                                                                                                                                                                                                                                                                                                                                  |
| 19          |                                                                                                                                                                                                                                                                                                                                                                                  |
| 20          |                                                                                                                                                                                                                                                                                                                                                                                  |
| 20          |                                                                                                                                                                                                                                                                                                                                                                                  |
| 21          |                                                                                                                                                                                                                                                                                                                                                                                  |
| 22          |                                                                                                                                                                                                                                                                                                                                                                                  |
|             |                                                                                                                                                                                                                                                                                                                                                                                  |
| 23          |                                                                                                                                                                                                                                                                                                                                                                                  |
| 24          |                                                                                                                                                                                                                                                                                                                                                                                  |
|             |                                                                                                                                                                                                                                                                                                                                                                                  |
| 25          |                                                                                                                                                                                                                                                                                                                                                                                  |
| 26          |                                                                                                                                                                                                                                                                                                                                                                                  |
|             |                                                                                                                                                                                                                                                                                                                                                                                  |
| 27          |                                                                                                                                                                                                                                                                                                                                                                                  |

# 28 Abstract

- Central Asia is a landlocked region with its freshwater resources originating in the mountains of Pamir,
- Tianshan, and Hindukush. Water resources in this area are formed mainly due to seasonal snowmelt,
- with glacier melt being the second largest hydrological component contributing to river flow, primarily
- in late summer. Water resources are shared among all Central Asian countries and used mainly for
- agricultural production purposes as well as hydropower generation. Proper management of water
- resources requires an accurate assessment of water availability originating in the mountains, mainly due
- to snowmelt. This requires data on snow depth, which is limited in the region. Snow surveys that were
- initiated during the 1980s have not continued in many parts of the region. The limitation of data on snow
- depth observations creates a challenge in forecasting water availability with the required accuracy.
- In order to cope with the challenge of data availability on snow depth measurements to improve the
- accuracy of hydrological forecasts, we introduced a novel approach that involved communities living in
- the source area of water formation to collect snow depth measurements. The project was conducted in
- the territory of Kyrgyzstan, Tajikistan, and Uzbekistan, and more than 1000 observations were collected
- in the period from February 2024 to March 2025. Figures and maps prepared for this manuscript rely on
- data collected in 2024. The social media channel Telegram was used to establish communication with
- communities living in remote areas. The observations were done voluntarily. Volunteers used a ruler as
- a measuring device and Telegram to send their observations every five days in the period of January to
- March 2024. The data on snow measurement were validated for any outliers by comparing them to the
- closest observations that were provided by other volunteers.
- The data collected in this project were used as ground-truth data to validate MODIS snow cover data
- that was processed by the MODSNOW-Tool. The validation results showed over 80 % agreement of
- community-driven snow depth measurement and snow cover observation from remote sensing products.
- In summary, community-driven snow depth data collection enhances the accuracy of mountain snow
- storage assessments, supports water resource forecasting, and fosters long-term resilience by
- empowering local participation in environmental monitoring, particularly valuable in resource-limited,
- remote regions like Central Asia.
- The dataset is freely accessible from https://doi.org/10.5281/zenodo.17158864 (last access: 19
- September 2025; Gafurov et al., 2025).
- **Keywords:** Central Asia, water resources, snow depth measurement, community involvement, citizen
- science

78

melt models.

# 1 Introduction

Ice and snowmelt are essential water sources for the densely populated countries in Central Asia and 68 play a critical role in the livelihoods of both lowland and mountain communities (Schaner et al., 2012; 69 Chen et al., 2016; Barandun et al., 2020; Didovets et al., 2021; De Keyser et al., 2023). In areas where 70 irrigation is widely practiced, these water resources create ongoing socio-hydrological interactions 71 (Thapa et al., 2021; Chen et al., 2024 Orazaliev et al., 2024). Snow accumulation in mountainous areas 72 primarily during the winter months serves as a natural reservoir, regulating river flow in spring and early 73 summer. In Central Asia, seasonal snow cover accounts for a significant portion of the annual water 74 supply, with snow melt contributions estimated to exceed 50% in major basins (Hoelzle et al., 2017; 75 Fallah et al., 2024). Snowmelt is the dominant contributor to annual runoff across the Tianshan (Aizen 76 et al. 1995). Armstrong et al. (2017) reported seasonal snow contributions as high as 65-72% of the 77 average annual runoff in the Amu Darya and Syr Darya basins using remote sensing and degree-day

Snow, glaciers, and permafrost serve as indicators of atmospheric fluctuations and highlight ongoing 80 environmental changes (Chen et al., 2024; Fallah et al., 2024; Wang et al., 2024). As climate change 81 progresses, the impact of altered meltwater from snow, ice, and permafrost will become increasingly 82 significant for the fragile mountain and lowland ecosystems of Central Asia (Varis, 2014; Hoelzle et al., 83 2017; Haag et al., 2019; Didovets et al., 2021; Kraaijenbrink et al., 2021; Chen et al., 2024; Fallah et al., 84 2024; Saidaliyeva et al., 2024; Wang et al., 2024). These shifts will impact livelihoods, particularly for 85 mountain communities, as well as densely populated downstream regions (Saidaliyeva et al., 2024). A 86 declining cryosphere could trigger severe ecological changes, jeopardizing water, food, and health 87 security, potentially leading to political instability and altered socio-hydrological dynamics (Acharya et 88 al., 2023; Fallah et al., 2024; Thapa et al., 2021; Saidaliyeva et al., 2024). Effective mitigation requires 89 enhancing adaptation capacity, starting with the creation of fundamental observational data on the 90 cryosphere's state and changes, followed by the development of well-calibrated models integrated with 91 climate scenario projections, helping to reduce the vulnerability of local populations in the mid- to longterm (Varis 2014; Barandun et al., 2020; Thapa et al., 2021; Acharya et al., 2023; Saidaliyeva et al., 92 93 2024).

However, our understanding of snow cover magnitudes, dynamics, and regional climate norms remains 95 limited due to the remoteness of the terrain, limited in situ observations, and the often-poor performance 96 of models (Barandun et al., 2020; Acharya et al., 2023). Precipitation patterns at high altitudes in Central 97 Asia are particularly uncertain; in situ observations are rare and often unreliable, especially at higher 98 elevations, resulting in significant uncertainties in observational datasets (Xu et al., 2018; Immerzeel et 99 al., 2015; Fiddes et al., 2019). Remote sensing methods (e.g., TRMM, CHIRPS) also tend to 100 underestimate solid precipitation, and their coarse grid scales introduce inaccuracies in complex terrains 101 (Immerzeel et al., 2015; Fiddes et al., 2019). Purely model-based approaches, such as the ERA5 reanalysis from the European Centre for Medium-Range Weather Forecasts, in which precipitation is 102 103 predicted rather than assimilated, have shown considerable biases over the High Mountain Asia region 104 (Barandun et al., 2020).

Collecting snow depth data is crucial for predicting water availability in Central Asia, particularly in countries such as Kyrgyzstan, Tajikistan, and Uzbekistan, where snowmelt is a vital water source for the agriculture and hydropower sectors. Traditionally, hydrological stations have provided this data, but they often face limitations in coverage and lack real-time information, especially in remote or high-altitude areas (Immerzeel et al., 2015; Fiddes et al., 2019). Timely snow data is necessary to enhance the accuracy of water resources and water availability forecasts, which are essential for water resource

- management, agricultural planning, hydropower generation, and overall socio-economic stability in these regions. Central Asian areas also suffered from a declining number of observational stations after
- the college of the Societa Union which further contributed to date one in coording amount of the College of the Societa Union which further contributed to date one in coording amount of the College of the Societa Union which further contributed to date one in coording amount of the College of the Societa Union which further contributed to date one in coording amount of the College of the Societa Union which further contributed to date one in coording amount of the College of the Co
- the collapse of the Soviet Union, which further contributed to a data gap in assessing snow storage and
- water availability.
- To address these challenges, this study introduces an innovative, community-driven approach to snow 116 depth data collection. Rather than relying solely on official meteorological stations, it leverages citizen 117 science by engaging local communities in the process. Citizen science is a collaborative approach to 118 scientific research that involves non-professional participants, often members of the public, in 119 contributing to various stages of the scientific process. This participation can include data collection, 120 analysis, interpretation, and even project design. Citizen science is widely applied in disciplines like 121 environmental science, ecology, astronomy, and public health, where it enables the generation of large-122 scale datasets, fosters public engagement with science, and provides opportunities for education and 123 community involvement. It bridges the gap between professional researchers and the public, 124 empowering individuals to actively contribute to addressing scientific and societal challenges (Conrad
- 425 & Hilchey, 2011; Fraisl et al., 2022). Citizen science has emerged as a powerful approach for
- environmental monitoring, leveraging the contributions of non-professional participants to enhance data
- collection and foster community engagement. However, successful implementation requires addressing
- challenges such as participant recruitment and retention, as well as data quality assurance.
- Ensuring data quality is a critical challenge in citizen science, where non-professional participants
- collect, analyze, or report data to support scientific research (Balázs et al., 2021). This paper explores
- the role of citizen science in cryosphere monitoring, examining its applications, benefits, and limitations.
- Specifically, it demonstrates the feasibility of collecting snow depth data in remote mountainous regions
- of Central Asia by engaging local communities to improve water resource predictions in areas with
- limited official observations. This study was piloted in the territory of Kyrgyzstan, Tajikistan, and
- Uzbekistan. Volunteers were involved in taking regular snow depth measurements and submitting the
- data, along with geolocation and photos, via the Telegram messenger mobile application. By engaging
- residents, the project addresses data gaps in remote or under-monitored regions, enabling a more
- comprehensive understanding of snow depth data and snow distribution. Community involvement also
- promotes local participation in environmental monitoring and resource management, helping to ensure
- the long-term sustainability of these data collection efforts.

# 2 Study Area

- The study area covers the mountainous regions of Central Asia, specifically focusing on the countries
- of Kyrgyzstan, Tajikistan, and Uzbekistan (Fig. 1). These regions are characterized by various
- landscapes such as vast mountain ranges, high-altitude plateaus, and dry lowlands. These distinct
- geographic characteristics are vital in the hydrological cycle and govern water resource availability at
- both local and regional levels (Tian et al., 2023). Snowmelt from these mountainous regions plays a key
- role in replenishing rivers, essential for both agricultural production and hydropower generation
- (Didovets et al., 2021). This highlights the importance of mountain snowmelt to the water resources and
- economy in these countries.

Figure 1. Study Area: Kyrgyzstan, Tajikistan, and Uzbekistan. Basemap data sourced from Natural Earth (public domain); all subsequent maps in the study use the same basemap.

Uzbekistan is an extensive and predominantly desert country. Its topography includes the vast Kyzylkum Desert and the Pamir and Tianshan Mountain foothills. Notably, the country's primary rivers, the Amu Darya and Syr Darya, originating in the Pamir and Tianshan Mountain ranges, are important for agriculture, water supply in the region, and for the Aral Sea (Belolipov et al., 2013).

Tajikistan and Kyrgyzstan are both known for their large share of the Tianshan and Pamir Mountain ranges. Tajikistan includes a large part of the Pamir Mountains and the eastern part of the Tianshan (Metrak et al., 2015). At the same time, over eighty percent of Kyrgyzstan's territory is covered by the Tianshan Mountains, and only the southern part of the Pamirs (Chotaeva, 2021). The high-altitude mountainous regions of Tajikistan and Kyrgyzstan are crucial for the accumulation of snow, which feeds important rivers such as the Amudarya and Syrdarya rivers.

# 3 Data and methods

## 3.1 Community Involvement

Snow depth measurements were taken by volunteers using a simple ruler to measure the depth in centimeters. These measurements were recorded every five days, ensuring a consistent data collection schedule throughout winter. The chosen interval allowed for frequent monitoring to capture changes in snow depth while being manageable for participants. Volunteers were crucial in this process, as they gathered and reported the measurements. Their involvement was essential for expanding data collection across vast and hard-to-reach regions, particularly in mountainous areas where traditional meteorological stations were scarce or nonexistent. This community-based approach significantly increased data coverage and ensured a broad geographic representation in the areas of data collection.

188

192

Volunteers were selected based on their geographical distribution to ensure diverse and relevant coverage, with special attention given to participants from mountainous areas, as snowmelt in these regions is essential for predicting water availability. A survey was conducted to identify participants from various altitudes, including those residing in foothills (700-1200 meters), mid-mountainous regions 176 (1200-2200 meters), and high mountains (above 2200 meters) by initiating a voting round in the Telegram Group called UZB-METEO, which is a dedicated group of an environmental blogger and has around 12,000 subscribers. This targeted recruitment ensured data from a wide range of elevations, 180 which is critical for snowmelt and water availability predictions.

#### 181 3.2 Data description

Each snow depth measurement was accompanied by recording the precise location using the geolocation 182 function through volunteers' smartphones. This was facilitated through the dedicated Telegram 183 184 messenger application that was prepared for this project.

Volunteers used the Telegram messenger app to submit their snow depth measurements, geolocation information, and photographic evidence of the measurements. The inclusion of photos ensured data accuracy by providing visual confirmation of the snow depth readings and allowed for further verification if needed. This streamlined data collection process enabled easy and efficient communication between volunteers and project coordinators, ensuring real-time updates and minimizing the risk of errors or inconsistencies in the data.

Figure 2. Images sent by volunteers displaying the snow depth measuring process. Used with permission from Erkin Abdulahatov, UZB-METEO group administrator.

# 3.3 Measurement Process

The snow depth measurement process followed a standardized protocol to ensure consistency and reliability. Participants were instructed to measure the snow depth using a ruler, recording the depth in centimeters. These measurements were taken every five days, ensuring regular monitoring throughout the data collection period. To maintain accuracy and data validity, volunteers were required to photograph the ruler in the snow, providing visual confirmation of the measurements. Additionally, each measurement's geolocation was recorded using the GPS functionality on their smartphones. Participants were given additional instructions on selecting representative locations for measurements, such as avoiding areas directly near houses or in sheltered locations and refraining from measuring in areas with unusually deep snow accumulation.

All data, including snow depth, geolocation, and photos, were submitted to the project's dedicated Telegram bot. This process allowed for real-time data submission, enabling project coordinators to

- quickly verify the accuracy of the measurements by cross-referencing the reported data with the provided photos.
- Once submitted, the snow depth data and the corresponding geolocation information were compiled into
- a central database. This compilation process involved organizing the data based on the geographic
- coordinates of each measurement, enabling the creation of detailed snow depth maps for the regions
- under study. These maps visually represented snow distribution across Kyrgyzstan, Tajikistan, and
- Uzbekistan. The geolocated data points were used to interpolate snow depths between measurements,
- offering insights into broader snow coverage in the areas where volunteers were active.

## 3.4 Data Visualization and Interpolation

- The resulting snow depth maps were updated regularly to reflect the collected data, helping researchers
- analyze snow accumulation patterns and forecast water availability based on the snowmelt potential in
- key regions.

214

- The recorded snow depth data was further interpolated and visualized using the Quantum GIS program
- to provide a comprehensive view of snow distribution across Kyrgyzstan, Tajikistan, and Uzbekistan.
- Geolocation data from each measurement point was processed and used to generate maps depicting
- snow depth variations across different altitudes and areas. The Inverse Distance Weighting (IDW)
- interpolation technique was employed to estimate snow depth in places where no direct measurements
- were taken, allowing for a continuous visual representation of snow cover. The IDW methodology,
- however, does not consider the terrain in the region and might thus be biased in areas with highly varying
- elevation ranges.
- The maps utilized different color codes to represent varying snow depth ranges in centimeters, making
- it easy to interpret the data visually. Observed snow depth points were marked with specific values,
- while the regions were color-coded to reflect the interpolated snow depths.

#### 229 3.5 MODIS Validation

- Data collection was conducted in the period from January to March 2024, which is a critical time for
- snow accumulation in Central Asia. This time frame allowed the project to capture the seasonal snow
- accumulation and its potential impact on water availability, offering valuable insights for forecasting
- and resource management.
- The collected data was used as ground-truth data for validation of (Moderate Resolution Imaging
- Spectroradiometer) snow cover data (Hall et al., 2003). The MODIS data was processed by the
- MODSNOW-Tool (Gafurov et al., 2016), including cloud removal. The validation was done by
- comparing the snow depth collected with the help of the community and the corresponding pixel
- covering the location of the community-driven point observation. As for the performance measure, the
- matching ratio between community-driven snow depth observation and MODIS snow cover was used
- (Eq. 1)

$$S_{eff} = \frac{N_{total}}{N_{true}}$$
 (1)

- where,
- $S_{eff}$  performance measure of comparison between community-driven snow depth observation
- and MODIS snow cover data
- $N_{total}$  total number of comparisons of community-driven snow depth observation and MODIS
- snow cover data

- $N_{true}$  number of true comparisons, where both community-driven and MODIS snow cover
- show agreement.

- For the performance measure estimation, only a binary type of comparison was done, e.g., snow
- covered (snow depth>0) or not snow covered (snow depth=0).

#### 4 Results

- The community-driven snow depth measurement project produced data spanning over several collection
- periods, capturing the variations in snow depths across Kyrgyzstan, Tajikistan, and Uzbekistan over
- different time frames. Snow depth observations were made within a 5-day interval, or in some cases,
- when fresh snowfall was observed. Throughout each observation day, the number of observations
- changed slightly due to the availability and engagement of the volunteers, as well as external or logistical
- changed signify due to the availability and engagement of the volunteers, as went as external of logistical
- constraints. Ultimately, the data collected through this community effort was compiled to create detailed
- observed and interpolated snow depth maps.
- In the following section, key findings for snow depth observations and interpolations over four key dates
- in the project are presented, while other significant findings are detailed in the supplementary document.
- These results are important for developing a more comprehensive understanding of snowpack dynamics,
- its impact on water resource management, and forecasting water availability in these regions.
- Notably, each map shows a gradient from light green to dark blue, representing interpolated snow
- depths. Light green indicates snow depths of less than 7 cm, medium shades indicate snow depths from
- 8 cm to 38 cm, and dark blue indicates snow depths exceeding 45 cm. These maps highlight the temporal
- and spatial dynamics of snow coverage across the study regions.
- 4.1 Community-Driven Snow Depth Observations and Interpolation

Figure 3. Snow depth observations and interpolations for Kyrgyzstan, Tajikistan, and Uzbekistan, with a total of 39 observations taken for January 31, 2024.

On January 31 (Fig. 3), initial observations revealed significant snow accumulations in Kyrgyzstan, with a depth of 121 cm recorded at Too-Ashuu Pass and 90 cm in the Akterek Village. The western, eastern, and southern parts of Kyrgyzstan show less snow accumulation, with 1 cm of snow recorded in the Kyzyl-Adyr Village. Snow accumulation in southern Uzbekistan also reached over 45 cm, with the Shahrisabz district measuring 94 cm and 48 cm in the Qamashi district, respectively. In contrast, much of Tajikistan showed lower snow depths, particularly in the eastern regions. The highest recorded snow depth in Tajikistan was 39 cm near Sanglok, part of the larger Pamir Mountain range, followed by 37 cm in the Tavildara Village.

Figure 4. Snow depth observations and interpolations for Kyrgyzstan, Tajikistan, and Uzbekistan, with a total of 44 observations taken for February 17, 2024.

Compared to January 31 readings, similar areas recorded higher snow depth values on February 7 (see Fig. S1). However, by February 17 (Fig. 4), central and western Uzbekistan experienced reduced snow coverage, with many areas reporting depths of 0 cm. In the northeastern regions, depths ranged from 0 cm to 82 cm. Moreover, Kyrgyzstan continued to show high snow accumulation, with depths of 114 cm at Too-Ashuu Pass. In contrast, most of Tajikistan recorded low snow depths, although the Shahristan district in the northeastern part of the country reported a snow depth of 27 cm.

Figure 5. Snow depth observations and interpolations for Kyrgyzstan, Tajikistan, and Uzbekistan, with a total of 62 observations taken for March 5, 2024.

On March 5 (Fig. 5), snow depths remained high in the mountainous regions, with the northeastern parts of Uzbekistan reporting up to 121 cm, and Anzob Pass in Tajikistan reaching 147 cm. Kyrgyzstan showed substantial accumulation, particularly in the Tianshan Mountain range. Notably, in Kyrgyzstan, snow depth patterns compared with other dates (see Figs. S2–S5) were largely consistent with each other, indicating steady temperatures, snow accumulation, and volunteer engagement.

Figure 6. Snow depth observations and interpolations for Kyrgyzstan, Tajikistan, and Uzbekistan, with a total of 73 observations taken for March 20, 2024.

By March 20 (Fig. 6), snow depths decreased in most regions, especially in central and southern Uzbekistan, northern and southern Tajikistan, and northwestern and southern Kyrgyzstan, where snow depths were primarily 0 cm. However, significant snow accumulation persisted in the mountainous regions. In Uzbekistan, a record depth of 111 cm was observed at the Maydontol station. Too-Ashuu Pass in Kyrgyzstan reached 129 cm, while Anzob Pass in Tajikistan recorded 160 cm, the highest snow depth of the study period.

Thus, the variations in snow depth underscore the role of elevation and local climate in snowpack determination, with the mountainous regions consistently observed with higher snow accumulations.

# 4.2 Snow depth changes over time at single observation points

Figures 7, 8, and 9 summarize snow depth changes over time at single observation points across different regions in Kyrgyzstan, Tajikistan, and Uzbekistan, respectively. Each graph captures the regional trends and highlights fluctuations in snow depth during the study period.

Figure 7. Snow depth measurements over time at single observation points across different regions in Uzbekistan.

Figure 8. Snow depth measurements over time at single observation points across different regions in Tajikistan. Note: Data for Anzob Pass (blue line) is recorded only on specific observation days.

315

312

Figure 9. Snow depth measurements over time at single observation points across different regions in Kyrgyzstan.

In Uzbekistan, data gaps in snow depth measurements were observed due to a lack of volunteer submissions. Nevertheless, Fig. 7 captures notable fluctuations in snow depths across the different regions. For instance, the snow depths in the Shahrisabz District in Qashqadarya peaked in late January and gradually decreased into late February and March. Snow depth observations for the Maydontol station in the Tashkent region were present in late February and peaked around March 14.

Fig. 8 shows that most regions in Tajikistan had continuous snow depth data without gaps. In Sanglok, snow depths gradually increased in February, reached over 80 cm in early March, and decreased later in the month. In comparison, the Shahristan District had consistently lower and more stable snow depths throughout the study period. In contrast, snow depth gaps were evident for Anzob Pass, where only four days of observations were made. However, a peak depth of 160 cm was recorded in late March.

Similar snow evolution dynamics were observed in Kyrgyzstan, where continuous snow depth measurements were reported across the regions (Fig. 9). Too-Ashuu Pass consistently maintained high snow depths of around 120 cm, with minimal variation, while areas like the Kyzyl-Adyr Village had much lower and stable snow depths, staying close to zero with only minor data gaps.

#### 4.3 Validation analysis for the Naryn and Karadarya River Basins

To assess the reliability of MODSNOW satellite-derived snow cover data on a localized scale, we conducted a validation analysis using community-observed snow depth measurements from two distinct catchments in Kyrgyzstan: Naryn and Karadarya.

In the Naryn catchment area, a total of 51 observations were recorded between January 31, 2024, and March 30, 2024. Data was collected on 13 different days at 4 distinct measuring points. According to MODIS data, the snow coverage was present in all 51 observations. Looking at the observed snow depth data there are 9 observations which do not show snow cover at all, therefore having a snow depth of 0 cm. This means that 42 out of 51 times, the observation data matched the MODSNOW data, resulting in an agreement percentage of 82.35%.

In the Karadarya catchment area, 13 observations were conducted from February 7, 2024, to March 30, 2024, spanning 13 days at a single measuring point. The observed snow depth consistently showed no snow cover (0 cm) on all 13 days. According to MODSNOW satellite data, there was no snow coverage

355

359

on 11 days ("no"), while snow coverage was indicated on 2 days ("yes"). This means that the data matches 11 out of 13 times, resulting in a percentage agreement of 84.62%.

# Naryn and Karadarya Catchment Basins (20.02.24)

Figure 10. Snow cover area for the Naryn and Karadarya Catchment Basins in Kyrgyzstan taken for February 20, 2024.

## Naryn Catchment Basin (20.02.24)

Figure 11. Snow cover area for the Naryn Catchment Basin in Kyrgyzstan taken for February 20, 2024.

Overall, the validation analysis demonstrates a strong agreement between MODSNOW satellite-derived data and ground-based observations in both catchments, supporting the reliability of MODSNOW for monitoring snow cover in the region.

4.4 Regional Validation of MODIS and MODSNOW-Tool Cloud-Removal Across Six River Basins

This section presents a regional validation of MODIS snow cover data and its enhancement through the MODSNOW-Tool's cloud-removal algorithm. The goal is to assess the agreement between satellite-derived snow cover (with and without cloud removal) and community-driven ground observations across six major river basins.

## 4.4.1 Accuracy of Original MODIS Snow Cover Under Cloud Conditions

First, a quality check of the original MODIS snow cover data was conducted to assess its accuracy in detecting snow in the presence of cloud coverage. As an example, results are shown for February 26, 2024, which was used as the day to show the validation results between the original MODIS and MODSNOW data. The quality check was done for the following river basins: Ahangaran, Chirchik, Fergana Valley, Kashkadarya, Naryn, and Wachsh Darband.

Figure 12. Original MODIS Snow Cover Data with Cloud Coverage for February 26, 2024.

Within the original MODIS data, a total of 55 snow observations were identified across the river basins between January 31, 2024, and March 30, 2024. Data was collected on 12 different days at 28 distinct measuring points, distributed across the six river basins. The number of measuring points within each river basin is as follows: Ahangaran (4 points), Chirchik (6 points), Fergana Valley (6 points), Kashkadarya (6 points), Naryn (4 points), and Wachsh Darband (2 points). Notably, MODIS snow cover data were unavailable for 2 of the 14 original community-driven snow depth observation dates: February 7 and 20. Therefore, these dates were not used.

Observed snow depth data (ground-truth) with MODIS snow cover data, considering the presence of cloud cover, were then compared. On 40 out of 55 occasions, MODIS and ground-truth data agreed that snow was present. On 2 out of 55 occasions, MODIS indicated no snow, but the ground-truth data indicated snow, leading to a disagreement. On 5 out of 55 occasions, MODIS indicated snow, but the ground-truth data showed no snow, also resulting in a disagreement. Finally, on 8 out of 55 occasions, both MODIS and the ground-truth data agreed that there was no snow. Therefore, the accuracy of

MODIS snow cover in detecting snow under cloud cover conditions resulted in an agreement with community-driven observations percentage of 88%.

## 4.4.2 Accuracy of MODSNOW-Tool Processed Snow Cover with Cloud Removal

Secondly, a quality check was performed on the MODSNOW-Tool to evaluate the performance of its cloud removal algorithm applied to the MODIS-derived snow cover data. After processing with the cloud removal algorithm, a total of 185 snow observations were identified across the 28 measuring points distributed throughout the six river basins.

Figure 13. MODIS Snow Cover Data processed by MODSNOW-Tool Cloud Removal for February 26, 2024.

On 131 out of 185 occasions, MODSNOW cloud removal and ground-truth data agreed that snow was present. On 19 out of 185 occasions, MODSNOW cloud removal indicated no snow, but the ground-truth data indicated snow, leading to a disagreement. On 14 out of 185 occasions, MODSNOW cloud removal indicated snow, but the ground-truth data showed no snow, also resulting in a disagreement. Finally, on 21 out of 185 occasions, both MODSNOW cloud removal and the ground-truth data agreed that there was no snow. Therefore, the accuracy of MODIS snow cover data processed by using the MODSNOW-Tool cloud removal algorithm resulted in an agreement percentage with community-driven observations of 82%.

Overall, while the original MODIS data showed slightly higher agreement under limited conditions, the MODSNOW-Tool significantly expanded data coverage and maintained strong overall accuracy, demonstrating its value in enhancing operational snow cover monitoring across the region.

# 5 Discussion

Collecting snow depth data is vital for predicting water availability in Central Asia, where snowmelt is a key resource for agriculture and hydropower sectors. Traditional snow measuring stations often lack

- coverage, especially in remote or high-altitude areas. This study introduces a community-driven 410
- approach, engaging local volunteers in data collection to fill gaps in snow depth monitoring to further
- improve water resource predictions and foster long-term sustainability through local involvement.
- Collected snow depth observations demonstrate changes over time in the region, illustrating regional 412
- snow evolution dynamics and highlighting fluctuations in snow depth during the study period.
- Additionally, validation analysis of the snow cover area in the Naryn and Karadarya Catchment Basins
- in Kyrgyzstan, using MODSNOW data, indicates that the observed data is closely aligned with
- MODSNOW, achieving an agreement of over 82 % and 85 %, respectively. 416
- Engaging local volunteers to collect snow depth observations in the mountainous, remote regions of
- Central Asia has proven to be highly useful despite the challenges. One of the primary advantages is the
- ability to gather data from areas that are often under-monitored or completely inaccessible to traditional
- hydrological stations. These regions are critical for understanding snowmelt, which is a key water
- resource for agriculture and hydropower sectors in countries like Kyrgyzstan, Tajikistan, and
- Uzbekistan. By involving local communities, researchers can obtain real-time, on-the-ground data that
- would otherwise be difficult and expensive to collect.
- Another significant benefit is the increased spatial coverage provided by volunteer efforts. Snow
- measuring stations are typically limited in number and concentrated in more accessible areas, which
- creates significant gaps in snow depth data across large and high-altitude regions. Local volunteers help
- fill these gaps, providing a more complete and detailed understanding of snow distribution and snowmelt
- patterns. This can lead to improved predictions of water resources and water availability, which is
- essential for effective water resource management and agricultural planning.
- Furthermore, the involvement of community members fosters a sense of ownership and participation in
- environmental monitoring, which can enhance the sustainability of data collection efforts. By
- empowering local populations to contribute directly to scientific research, the initiative builds a network
- of engaged citizens who may also become more invested in broader issues related to water resource
- management and climate adaptation. This can help build resilience in these regions, where climate
- change poses an increasing threat to water security and livelihoods.
- The collection of community-driven snow depth measurements can also be cost-effective for
- hydrometeorological centers in Central Asia. These institutions used to collect snow depth information
- using helicopter flights to mountainous areas and by reading snow depth information from snow 438
- measuring stakes. Such stakes were installed during Soviet times in Central Asia. The spatial
- observations of these stakes (e.g., snow depth readings) occur on a single day, which requires a
- helicopter flight to a mountainous region. Helicopter services are expensive in Central Asia, and national
- hydrometeorological services do not always have the resources to conduct such costly observations.
- Therefore, collecting snow depth data with the help of communities is useful due to both a) being less
- costly and b) being continuous compared to spatial observations that take place only a few times (in
- some cases, only one time or none) during the winter period.

#### 6 Limitations

The validation analysis shows that the agreement of MODSNOW-tool data with community-driven 447

- observations is nearly 80%, resulting in nearly 20% of differences. One of the primary limitations of
- this community-based snow depth data collection project was low participation from high-elevation 449
- areas, where snow accumulation plays a critical role in determining water availability. While efforts
- were made to involve volunteers from mountainous regions, only 3% of 12,522 members of the
- Uzb\_Meteo channel were from high-altitude locations (above 2200 meters). The lack of adequate
- representation from these critical regions resulted in gaps in the data, particularly in areas most affected
- by snow accumulation and snowmelt.

- Additionally, some volunteers faced technological challenges that affected their ability to submit accurate and timely data. Many participants, especially in rural and remote areas, were unfamiliar with the mobile application Telegram that was used for data collection. This unfamiliarity led to difficulties in submitting geolocated data and photos, which could compromise the accuracy and reliability of the snow depth measurements. Some volunteers also struggled to send data in real-time in areas with limited internet connectivity, leading to delays or missing entries.
- Even though the project aimed to collect snow depth data every five days, this frequency was not always achieved. On some days, fewer submissions were received than expected due to participants facing challenges submitting data on time. In such cases, reminders were sent to volunteers, prompting them to submit their measurements the following day, which somewhat disrupted the intended schedule. Therefore, motivational strategies should be employed to ensure participants submit data promptly and remain engaged in the project.
- Alternative data collection methods were employed to address challenges related to infrequent data collection. Phone-based measurements were conducted in areas where participants could not effectively use the app. Volunteers were contacted via phone to provide their snow depth measurements, which were recorded manually by project coordinators. However, these phone-based submissions lacked geolocated photos for verification, which increased the potential for measurement errors and inconsistencies. While this method allowed for data collection in areas with limited technological access, it also introduced biases and reduced the overall accuracy of the dataset.

#### 7 Conclusion

The study marks the first significant effort to demonstrate the usefulness of community engagement for collecting snow depth data in the mountainous regions of Central Asia, a region where accurate and timely snow data is crucial for water resource forecasting. The initiative is particularly groundbreaking as it addresses a long-standing challenge in these remote areas: the lack of reliable, high-coverage snow depth observations due to the sparse distribution of hydrometeorological stations and the difficulties in accessing high-altitude terrains. Traditionally, snow depth data in Central Asia has been limited to a few official monitoring stations, often located in more accessible or lowland areas, leaving significant gaps in high-elevation zones where snowmelt is a primary water source. By involving residents in these mountainous and remote regions, this project not only fills these critical gaps but also provides an unprecedented level of detail and real-time data that can greatly improve water resource forecasts. This is especially important in Central Asia, where agriculture, energy production, and drinking water supplies rely heavily on the seasonal snowmelt.

Community engagement in this context goes beyond simply gathering snow measurements. It taps into the local knowledge of residents who are intimately familiar with the landscape and weather patterns, ensuring that data is collected from areas that would otherwise be overlooked. This bottom-up approach allows for the frequent and widespread gathering of snow depth data, which can then be aggregated to improve hydrological models and forecasting tools. These enhanced models are vital for predicting water availability, especially in regions where snowmelt serves as the primary source of freshwater for irrigation and household use during the dry months. Moreover, the success of this initiative has broader implications for resource management and environmental monitoring in the region. It sets a precedent for how citizen science can be effectively used to address data shortages in difficult-to-access areas, offering a scalable model that could be applied to other environmental parameters such as vegetation cover or air quality, which, however, might require additional measurement devices. The advantage of measuring snow depth using community support is that no costly device is necessary as a measuring

© Author(s) 2025. CC BY 4.0 License.

- tool. The methodology of snow depth collection using community support could be particularly
- beneficial in regions facing similar challenges of limited official monitoring infrastructure.
- The community-driven snow depth data collection can contribute to the improvement of water resource
- forecasts by enhancing and improving the quality of snow storage assessment in the mountains. It
- underscores the idea that involving the community can not only help solve immediate data collection
- problems but also build long-term resilience through local ownership of environmental monitoring
- processes. This pioneering approach offers valuable insights into how communities can play a critical
- role in supporting scientific efforts, especially in regions where resources and infrastructure are limited.
- While there are challenges associated with training, consistency, and technology use, the engagement
- of local volunteers in snow depth observations provides valuable data that significantly enhances
- scientific understanding of snow dynamics in remote, mountainous regions of Central Asia. This
- approach not only expands the reach of data collection but also strengthens community involvement in
- resource management and environmental protection.

# 512 Data availability

- All the data sets described and presented in this paper can be openly accessed at
- https://doi.org/10.5281/zenodo.17158864 (last access: 19 September 2025; Gafurov et al., 2025) and
- distributed under the CC BY 4.0 license.

#### 516 Supplement

The supplement related to this article is available online.

#### 518 Author contributions

- Abror Garfurov led the project and contributed to manuscript revision. Akmal Gafurov was responsible
- for MODIS data processing. FG and RA conducted the data collection. AS contributed to data analysis.
- EG contributed as editor. ZK contributed as a researcher. ZK, AS, and EG drafted the manuscript. FB
- contributed to paper editing. NA managed the IT infrastructure, while EA coordinated the Telegram
- group (Uzb\_Meteo). JK oversaw overall project management.

#### 524 Competing Interests

The contact author has declared that none of the authors has any competing interests.

#### 526 **Disclaimer**

# 527 Special issue statement

- This article is part of the special issue "Hydrometeorological data from mountain and alpine
- research catchments." It is not associated with a conference.

## 530 **Acknowledgement**

We thank all volunteers and the Uzb\_Meteo channel for their participation and support.

## 532 Financial support

- The data collection process is financed by Innovative Water and Environmental Solutions (IWES).
- Manuscript preparation is financed by GFZ Helmholtz Center for Geosciences and Innovative Water
- and Environment Solutions (IWES).

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
