# Peer review of "(untitled)"

_Earth System Science Data, 2025_

## Referee Comment (RC1)

Review ESSD-2025-580: **A novel approach: community-driven snow** 1 **depth measurement in Central Asia**

The authors present a dataset of snow depth observations from 3 Central Asian states collected by citizen scientists in 2024 and compare the data with remote approaches. Such CS work is very interesting and has a multitude of potential benefits but also challenging to set up, undertake and then document. That this has been attempted for the region is laudable and in principle of interest for the academic audience. However, there are a number of issues at this stage that currently suggest that this study isn't ready for publication in its current form. I summarize them below in 6 different sections (of which the last two are easier to solve). Since these issues would require substantial edits, I refrain from more detailed comments of the text (which in general is clearly written).

**Dataset and data documentation:** The ESSD Journal has a clear data submission and documentation policy, including specifically on spatial data. Datasets need to be accessible comprehensively (ideally in geopackage files, not individual shp files and definitely not xls files. Also a clear MetaData file is required that explains all data fields, associated units, uncertainties etc.

**Introduction and scope of the Journal:** The Introduction needs thorough revision. First, crucial statements on the relevance of snow are followed by inappropriate citations (often just a number of studies stringed together that do not relate to each other much), second the probably most important study on in situ snow observations (including citizen science) in the region is not mentioned at all (Bair et al., 2020), nor is any discussion of citizen science in the environmental context (which is relevant to the accuracy of the approach) attempted (but CS does exist in the region, if not extensively in the snow then in the hydrology domain and has been published).

L87f/L92f: The citations following the statement on ecological/social/political changes all seem misplaced, ranging from literature on avalanches to hydrology, none of which make any conclusions on social stability or ecology. Important to check and update this. Again in L92 most of these studies are not concerned with vulnerabilities. On the other hand the issue of hazards/avalanches isn't discussed in the Introduction, to which for example (Acharya et al., 2023) would be the relevant citation.

L130f: The statement following 'This paper explores the role of citizen science in cryosphere monitoring, examining its applications, benefits, and limitations...' is problematic in the context of this Journal. ESSD generally publishes datasets, where the focus is on the dataset, not the specific challenges as outlined in your statement above. If this is the intended focus, a Journal focusing on Citizen Science or the Cryosphere generally.

**Data Ethics**: In CS work there needs to be a clear statement on how the volunteers were informed what happens with their data and if they agreed to it. This remains missing.

**Quality Control and comparison to MODSNOW/MODIS:** While the study describes how data has been collected it does not go into any specifics on potential errors, uncertainties etc, which is crucial in any case, but all the more so for CS approaches. This also extends to checking the usefulness of the sites that were submitted. Just a quick check reveals many to be in urban settings (Figure 1), which isn't ideal for a snow measurement and definitely would not to be

expected to match up with any remote data. The validation with the MODSNOW results is also confusing as a comparison to snow depth is suggested in the Figures 10 and 11 but only whether there was any snow or no snow is discussed. Hence the depth aspect of the measurement becomes less useful or at least not assessed and the value of the MODSNOW tool is also questioned considering it shows lower performance than MODIS raw data itself.

[Figure]

*Figure 1: Two example locations from the dataset in urban terrain, where variability and human impact is expected to be large.*

**Methodology/Interpolation:**

As the authors themselves state, an IDW interpolation is problematic for snow, especially in complex terrain. It's unclear why it was performed then, considering that it adds no clear extra value here but would rather suggest snow cover where there may be none and vice versa. Why was not one of the more advanced tools used (like Kriging including the topography, (Kusch & Davy, 2022))?

**Title:** To me it is not clear where the novel is exactly coming from. Community-driven snow observations exist in the region since a while (see e.g., Bair et al., 2020) and the integration of citizen science data into 'conventional' data is also not new. I would advise to drop the word, or alternative make a strong case why this is novel exactly.

**Literature**

Acharya, A., Steiner, J. F., Walizada, K. M., Ali, S., Zakir, Z. H., Caiserman, A., & Watanabe, T. (2023). Review article: Snow and ice avalanches in high mountain Asia – scientific, local and indigenous knowledge. *Natural Hazards and Earth System Sciences*, *23*(7), 2569–2592. https://doi.org/10.5194/nhess-23-2569-2023

Bair, E. H., Rittger, K., Ahmad, J. A., & Chabot, D. (2020). Comparison of modeled snow properties in Afghanistan, Pakistan, and Tajikistan. *The Cryosphere*, *14*(1), 331–347. https://doi.org/10.5194/tc-14-331-2020

Kusch, E., & Davy, R. (2022). KrigR—a tool for downloading and statistically downscaling climate reanalysis data. *Environmental Research Letters*, *17*(2), 024005. https://doi.org/10.1088/1748-9326/ac48b3